# Emotional Competence Mediates the Relationship between Communication Problems and Reactive Externalizing Problems in Children with and without Developmental Language Disorder: A Longitudinal Study

**DOI:** 10.3390/ijerph17166008

**Published:** 2020-08-18

**Authors:** Neeltje P. van den Bedem, Julie E. Dockrell, Petra M. van Alphen, Carolien Rieffe

**Affiliations:** 1Developmental Psychology, Leiden University, Wassenaarseweg 52, 2333AK Leiden, The Netherlands; crieffe@fsw.leidenuniv.nl; 2Department of Psychology and Human Development, University College London, 25 Woburn Square, London WC1H 0AA, UK; j.dockrell@ucl.ac.uk; 3Royal Dutch Kentalis, Theerestraat 42, 5271 GD Sint-Michielsgestel, The Netherlands; pmvanalphen@gmail.com

**Keywords:** reactive aggression, proactive aggression, DLD, SLI, emotion regulation, emotion recognition, development, adolescence

## Abstract

Language problems are a risk factor for externalizing problems, but the developmental path remains unclear. Emotional competence may mediate the relationship, especially when externalizing problems are reactive in nature, such as in Oppositional Deviant Disorder (ODD) and reactive aggression. We examined the development of reactive and proactive externalizing problems in children with (*n* = 98) and without (*n* = 156) Developmental Language Disorder (DLD; age: 8–16 years) over 18 months. Relationships with communicative risk factors (structural, pragmatic and emotion communication) and the mediating role of emotional competence (emotion recognition and anger dysregulation) were examined. Multi-level analyses showed that increasing emotion recognition and decreasing anger dysregulation were longitudinally related to decreasing ODD symptoms in both groups, whereas anger dysregulation was related to more reactive aggression in children with DLD alone. Pragmatic and emotion communication problems were related to more reactive externalizing problems, but these relationships were mediated by emotional competence, suggesting that problems in emotional competence explain the communication problems of children with DLD. Therefore, in addition to interventions for communication skills, there is a need to address the emotional competence of children with DLD, as this decreases the risk for reactive externalizing problems.

## 1. Introduction

Approximately two children in every classroom experience significant language problems that are not explained by other disorders [1]. These children are eligible for a diagnosis of developmental language disorder (DLD) [2]. Children with DLD experience difficulties expressing their own thoughts and wishes through language, and often have misunderstandings with others. These communication problems may cause frustration and have negative affect, resulting in externalizing behavior problems, such as aggression or oppositional behavior. Indeed, higher levels of externalizing problems have been found in children with DLD [3,4,5,6]. Language problems not only have a direct effect on the development of externalizing problems [7], but may also impede the development of emotional competence development in children, which in turn puts children at risk of the development of externalizing problems [8,9]. Understanding the direct and indirect effects of language problems on the development of externalizing problems is important both to inform models of behavior difficulties and in order to help professionals recognize and target the underlying causes of the problems. Therefore, the current longitudinal study examined externalizing problems in children between 9 and 16 years old with and without a formal diagnosis of DLD. We examined direct risk factors (severity of communication problems) and indirect risk factors (problems in emotional competence) for the development of externalizing problems.

### 1.1. Externalizing Problems in Children with Developmental Language Disorder

DLD is a neuro-developmental disorder which causes significant problems in language development and severe difficulties using language in daily live. Children with DLD often experience problems with the structural aspects of expressive language (e.g., word finding problems, or difficulty in making grammatical sentences) and/or in the understanding of language (e.g., small lexicon, difficulty understanding complex phrases, or slow processing of language). Additionally, some children experience problems in the social use of language (pragmatics), such as ordering information to tell a story and the understanding of jokes. The communication problems of children with DLD are not explained by other neuro-developmental disorders, hearing loss, or intellectual disabilities [2,10]. Language problems are often present from an early age and persist as children became older [11]. However, language problems can also appear during middle school when the communicative demands of the environment increase [12,13].

In children and adolescents with DLD, elevated levels of externalizing problems have been reported both by parents and teachers ([3,4,5,6,14], although problems are often not in the clinical range [3,5,15,16] and not all children experience these difficulties. To date, two studies have examined the externalizing problems of children with DLD longitudinally [5,17]. Different developmental trajectories are reported by teachers on the Strengths and Difficulties Questionnaire (SDQ). One study found stable levels of externalizing problems from 10 to 12 years of age with increasing levels to the age of 16 [17], whereas the other study found decreasing levels of externalizing problems in children and adolescents between the age of 7 and 16, resulting in norm-like levels at the age of 16 [5]. However, the adolescents in the latter study reported higher levels of externalizing problems at the age of 16 compared to their peers without DLD on the self-report version of the SDQ [18]. The different patterns of results may reflect that different forms and functions of externalizing problems were not distinguished. 

### 1.2. Distinguishing Reactive and Proactive Externalizing Problems in Children with DLD

Externalizing problems can be categorized as reactive or proactive behaviors. Reactive externalizing behaviors include expressing anger, or harming others after provocation or goal thwarting. In contrast, proactive externalizing behaviors are typically not anger induced, but are more instrumental in nature, such as threatening or manipulating someone to gain something from that person or to gain social status [19]. Reactive and proactive externalizing problems often co-occur in children, but different antecedents and developmental routes have been distinguished [20,21]. Distinguishing these different externalizing problems may provide a clearer picture of the externalizing problems experienced by children and adolescents with DLD and provide an insight into the developmental role of language problems in different externalizing problems.

To date, only one small study (*N* = 12) has examined reactive externalizing problems in children with DLD between 8 and 12 years of age [14]. Teachers reported that children with DLD, in comparison to children without DLD, reacted more angrily or aggressively when provoked by their peers. To date, proactive aggression has not been examined in children with DLD. Some studies have examined bullying, rule-breaking, or delinquent behavior, but have found no differences between children with and without DLD [4,6,22,23]. However, 19-year-olds with DLD reported more criminal convictions than their peers without DLD [24], but the reasons for these convictions (reactive or proactive) were not specified. Overall, children with DLD are at risk of externalizing problems, but primarily for reactive externalizing problems.

There is significant variation in the level and development of externalizing problems reported within the group of children with DLD. Positive associations have been found between externalizing problems and the severity of pragmatic or expressive language problems [5,6,25], but other studies have found no associations with the severity of expressive and receptive language problems of children with DLD [4,17,18]. Therefore, other factors are likely to play a mediating role in the development of externalizing problems.

### 1.3. Emotional Competence as a Mediating Factor in the Development of Externalizing Problems in Children with DLD

In typically developing children, reactive externalizing problems are common in toddlers, but show a sharp decrease between 2 and 4 years of age [20]. This decrease has been linked to improved language abilities. When language abilities develop, children increasingly understand what is going to happen, in which order and why, which reduces frustration and provides children with a sense of security, diminishing externalizing problems. Additionally, oral language helps children to express wishes through words instead of physical actions [26] to gain positive social interactions with their parents and peers and to develop academically. These factors have all been found to diminish the risk of externalizing problems [27,28].

However, language development also positively affects children’s ability to understand and regulate their emotions [29], which in turn is related to lower levels of externalizing behaviors [30,31]. When children have difficulties recognizing others’ emotions and intentions, they can experience social interactions as hostile, resulting in more reactive aggressive or oppositional reactions [19,30]. Children gain an understanding of their own and other’s emotions, intentions and behaviors through social interactions with their parents and peers [32,33]. As language is an important prerequisite for the development of aspects of emotional competence [9,34], problems in emotional competence might mediate the relationships between language problems and externalizing problems in children with DLD.

The development of emotional competence is often delayed in children with DLD. They experience more difficulties recognizing their own and other’s emotions, regulating emotions, and communicating about emotions [32,33,35,36]. These problems in emotional competence in turn are important risk factors for the development of reactive externalizing problems in children without DLD [8].

Proactive aggression is typically not related to anger dysregulation, as children do not act out of spite or frustration, but rather act in a calculating manner [19,37]. Nevertheless, children who lack the skills to communicate about their own emotions may use more behavioral strategies to show others what they want and feel, resulting in more acts of proactive aggression [26,38].

### 1.4. Present Study

Overall, previous studies suggest that externalizing problems are more common in children and adolescents with DLD, although longitudinal studies provide mixed results regarding patterns over development. However, these studies did not differentiate between reactive and proactive externalizing problems. Therefore, the first aim of the current study was to examine longitudinally reactive (oppositional behavior and reactive aggression) and proactive externalizing problems (proactive aggression) of children with and without DLD between 8 and 16 years old across an 18-month period. We expected more reactive externalizing problems in children with DLD [14], but no differences in proactive problems [4,6,22].

Additionally, we wanted to explain the individual differences in externalizing problems between children with and without DLD and within children across time. Therefore, the second aim of the study was to examine longitudinally whether the level and development of emotional competence could explain individual differences in proactive and reactive externalizing problems across time. We expected that higher and increasing levels of emotional competence as children become older would be related to fewer and decreasing reactive externalizing problems [30,31], but would not reduce proactive problems [19,37]. Further, we explored whether the strengths of these relationships were similar in children with and without DLD. As children with DLD have more difficulties in developing their emotional competence, this may be a stronger risk factor for externalizing problems in these children.

The third aim was to examine the relation between the severity of communication problems and externalizing problems of children with and without DLD and to examine whether problems in emotional competence mediated this relationship in children with and without DLD. We expected that higher levels of communication problems in children with and without DLD would relate to more externalizing problems (Figure 1). We examined relationships with structural and pragmatic communication problems, but also with emotion communication problems, that is children’s difficulties communicating about their own emotions with others. We expected that these communication problems would relate to more reactive externalizing problems, but that the relationships would be mediated by problems in emotional competence (Figure 1) [9]. Additionally, we expected more proactive aggression in children with more communication problems ([26,38], but did not expect that emotional competence would mediate this relationship [37].

## 2. Materials and Methods

### 2.1. Design

The present study is part of a larger study on the social–emotional development of children with and without communication problems, due to DLD, hearing loss or autism. We examine the longitudinal relationships between children’s communicative and emotional competence and their social, internalizing, and externalizing problems. Children between 9 and 16 years old and their parents filled out questionnaires three times over an 18-month period with 9 months in between each measurement. The test sessions for children lasted between half an hour and an hour depending on the speed of the children [23,32,39,40]. The study received approval from the Ethical committee of Psychology at Leiden University, the Netherlands (project 1308277752).

The present study focusses on the externalizing problems of children with and without DLD. Children reported on their reactive and proactive aggression, while parents reported on the oppositional behavior and the emotional competence of their child on the three measurements. Additionally, during the second measurement, Performance IQ (PIQ) was tested and parents reported on the communication problems of their child.

### 2.2. Participants

An active consent procedure was used. Children and their parents were invited to participate in the study through their schools and through organizations which provide support to children with DLD in regular schools. The schools and organizations sent information letters to the children who were eligible for the study. Parents and the children above 12 years of age received information about the goals and procedure of the study and were asked to indicate whether they were willing to participate in the longitudinal study. Additionally, participants were asked for permission to use available PIQ information from school or medical files. Parents indicated whether their child had a formal diagnosis of DLD, ASD, hearing loss or anything else, which was verified in school or medical files for the DLD group. Children were invited to participate in the study if they had no diagnosis (control group) or if they only had a DLD diagnosis (DLD group). In the current study, we only report on children for whom parent reports were available.

A total of 254 children between 8 and 16 years old participated in the study, of whom 98 children had DLD (Table 1). In the Netherlands, children are eligible for a diagnosis of DLD when their language abilities are 2 SD below the mean on a general language measure or 1.5 SD below the mean on two out of four structural language areas. The Clinical Evaluation of Language Fundamentals [41] is typically used to test language abilities. Most children with DLD attended specialized schools for children with DLD (72%), where they received education in small classrooms with extra support for their language development and more visual support. The other children with DLD attended mainstream schools with specialized help within their schools (28%). Children typically have a counsellor who regularly visits the school to advise the teacher and support the child in school work and social issues.

Children without DLD were included when they did not have any diagnosis as indicated by their parents and when their language abilities and performance IQ (PIQ) were within the normal range (95% Confidence Interval [CI] of a score of 85 or higher) as tested with two subtests of the Dutch version of the Clinical Evaluation of Language Fundamentals and two subtests of the Wechsler Intelligence Scale for Children (WISC) [42].

The majority of the questionnaires were filled out by one or two of the parents of the children (without DLD: 100%; with DLD: 93%). In 7 percent of cases, the DLD group questionnaires were filled out by someone else (teacher, brother or not further specified). The majority of the parents originated from the Netherlands (DLD: 92.9 %; without DLD: 93.5%). The other parents originated from Morocco, Turkey, Surinam, Britain or other unspecified countries. We do not have information on the language which the parents used at home with their child.

### 2.3. Materials

Externalizing problems were measured with the ODD scale of the Child Symptom Inventory (CSI) [43], which measures whether the child often argues with and disobeys the parent, is easily mad at or annoyed by others, blames others when something goes wrong and wants to take revenge, often has anger tantrums, and purposefully tries to annoy others. Parents indicated on a 4-point Likert scale how often the behavior occurred.

Children reported on their reactive and proactive aggressive behavior using the Instrument for Reactive and Proactive Aggression Self-report (IRPA) [44]. Children were presented with five different aggressive behaviors (hitting, pushing, kicking, scolding, or picking a fight) with three reactive motivations: because I was mad, because I was being bullied, or because I was scolded; and three proactive reasons: because I wanted to be mean, because I thought it was fun, or because I wanted to be the boss. Children reported for each motivation whether they had performed the aggressive act (almost) never (1), sometimes (2), or often (3). If children did not perform the aggressive act, they reported never on every question. The validity and reliability of the CSI and the IRPA were good [43,44], as was the reliability in our study for children with and without DLD (α > 0.79; Table 2). Mean scores were obtained for all scales.

Emotional competence was measured with the Emotion Expression Questionnaire [45]. Parents indicated how often their child correctly recognized the emotions of others (emotion recognition), and how often, how long and how strongly children expressed their anger (anger dysregulation) on a 5-point Likert scale. Both scales have acceptable reliability (α > 0.72) [45], as was found in our study for children with and without DLD (α > 0.75; Table 2). Mean scores were obtained for both scales.

Communication problems were measured with the Child Communication Checklist-second edition (CCC) [46,47]. Parents rated how often their child experiences problems in four structural language areas (speech, syntax, semantics, and coherence) and four pragmatic language areas (initiation of conversations, non-verbal communication, use of context, and stereotypical language use). The sum of the final four scales gives the pragmatic problems scale, while all scales combined give an indication of the severity of the general communication problems of a child. Standardized scores are available for the Dutch population. The general communication problems and pragmatic scales are reliable in children with and without DLD, as was found in our study (*α* > 0.83; Table 2). However, the separate structural scales are not reliable in children without DLD [47] and were only examined in children with DLD. There were missing data for six children with DLD and 13 children without DLD, due to a non-response of the parents or invalid filled-out questionnaires. These children were excluded from the analyses where the CCC was used.

Additionally, we examined problems children experience when communicating about emotions with the Children Alexithymia Measure (CAM) [48]. Parents rated whether their child had difficulties in communicating their own emotions, deflected attempts to talk about emotions, or said that they were fine while they seemed not to be. Parents reported on a 4-point Likert scale how often problems occurred. As in the validation study [48], good reliability was found for both groups (α > 0.91; Table 2). Mean scores were obtained. There were missing data for 11 children with DLD and four children without DLD due to a non-response of the parents. These children were excluded from the analyses where the CAM was used.

When data were not available from school or medical files, PIQ was measured with two subtests of the WISC [42], namely block design and picture arrangement, which are highly correlated with a complete IQ test (*r:* 0.79) [49]. Eight children with DLD and all children without DLD were tested during the second assessment point. Data were missing for six children with DLD and ten children without DLD due to attrition, or because we did not obtain parental permission to test the PIQ.

### 2.4. Procedure

Children were tested in a quiet room by a test leader who had received extensive training. We used a detailed protocol for the test session in order to provide the same instructions to participants. At the start of the test session it was stressed that there were no right or wrong answers and that answers were anonymous. Participants could read the questions and privately answer options on a laptop or tablet. For children with DLD, everything was also read aloud. Parents filled out anonymized questionnaires about their child on paper or via the internet. The externalizing problems and emotional competence scales were filled out at all three measurements with nine months in between, whereas the communication problems and PIQ were measured once during the second measurement.

### 2.5. Statistical Analyses

Our first aim was to examine the mean level and development of different externalizing problems and emotional competence of children with and without DLD. We fitted multi-level models using R (version 3.3.2) [50] to account for the dependency in the longitudinal data. In multi-level models, all available data points of a child are included in the analyses. First, we fitted a basic means model and entered age and the control variables (gender, SES, and PIQ) one at the time. Control variables were only kept in the model when they provided a better model fit (as indicated by a significantly lower Akaike’s Information Criterion (AIC)). Next, in order to compare the mean levels of externalizing problems of children with and without DLD, diagnosis was added to the model. Additionally, we added the diagnosis x age interaction in order to compare the mean levels of both groups across time. Predictor variables are significant when zero is not in the 95% Confidence Interval (CI). All models were fitted with the addition of a random slope, but this did not provide a better model fit and was excluded. We used a bootstrap procedure with 5000 bootstrap samples as a robust procedure to deal with non-normally distributed data [51].

The second aim was to examine whether individual differences in externalizing problems could be explained by children’s level and the development of emotional competence. We examined whether between-person differences in emotional competence explained their level of externalizing problems. Therefore, the mean level (of the three measurements) of emotion recognition and anger dysregulation were added to the model. Additionally, the longitudinal data enabled us to examine whether within-person increases in emotional competence related to decreasing levels of externalizing problems across time. Therefore, we added person specific change scores for every time point on emotion recognition and anger dysregulation (Time1—mean, Time2—mean, and Time3—mean) which reflect the changes in emotional competence within individuals across time [52]. Further, we examined whether the relationships between emotional competence and externalizing problems were similar in children with and without DLD by adding the interaction terms of diagnosis x emotion recognition (mean and change) and diagnosis x anger dysregulation (mean and change). Non-significant predictors were excluded from the model.

The third aim was to examine whether the relationships between externalizing problems and communication problems were mediated by children’s emotional competence (Figure 1). First, we examined the direct path of communication problems to externalizing problems. We reran the best fitting models as described above excluding children with missing data on the CCC or CAM. Then the severity of communication problems (general, pragmatic, or emotion communication), as well as the interaction effects with diagnosis, were added to the model. Second, we examined the direct path of communication problems to emotional competence. Third, we examined the indirect route of communication problems through emotional competence to reactive externalizing problems. Therefore, we added the communication problem scales to the analyses with emotional competence. When the communication problems no longer contribute to the model when emotional competence is added to the model, this suggests mediation, which was tested using a direct test of mediation with 10,000 clustered bootstraps [53].

## 3. Results

### 3.1. Preliminary Analyses

The groups with and without DLD were comparable in age (*t*(160.05) = 0.27, *p* = 0.801) and gender distribution (*Χ*^2^(1) = 0.405, *p* = 0.604). Children with DLD had a lower PIQ than their peers without DLD (*t*(228.28) = 7.91, *p* < 0.001), and lived in neighborhoods with a lower socio-economic status (SES) as indicated by their postal code (*t*(251) = 4.57, *p* < 0.001) (Table 1). However, the mean neighborhood SES of children with DLD was similar to the mean level of SES in the Netherlands, whereas the SES of children without DLD was slightly above the mean. SES and PIQ were controlled in the analyses.

As in all longitudinal studies, we had some attrition. Eleven children stopped after the first measurement, leaving 79 children with DLD and 155 children without DLD. After the second measurement, another 49 children did not continue leaving 62 children with DLD and 132 children without DLD. Children without DLD who did not participate throughout the three waves had a lower SES status than children who participated at all three measurements (*U*: 1417.00, *z*: −3.37, *p* = 0.001). In the DLD group, no differences were found on any of the study variables. All available data points of the participants were used in the multi-level models.

There were missing data of the communication problems questionnaires. It was tested whether these missing data were random. Children without DLD with missing CCC data were older and had lower SES compared to children with CCC data (*U*: 1328.50, *z*: −2.66, *p* = 0.008, and *U*: 1473.50, *z*: −2.02, *p* = 0.044 respectively) and children with DLD without CCC data had lower PIQ compared to children with CCC data (*U*: 112.50, *z*: −2.32, *p* = 0.020). However, no differences were found on emotional competence and externalizing problems. Children in both groups with missing CAM data had lower SES than children for whom data were available (Without DLD: *U*: 1111.00, *z*: −2.29, *p* = 0.022; DLD: *U*: 341.50, *z*: −2.77, *p* = 0.006). There were no differences on any of the other study variables.

### 3.2. Group Differences

The level and development of externalizing problems and emotional competence were compared between children with and without DLD. Table 2 shows the grand means of all study variables and Figure 2 shows the raw data and mean level across time for children with and without DLD. The self-reported levels of aggression and the parent-reported ODD symptoms were generally low in both groups. Approximately half of the children with and without DLD never reported a proactive act of aggression. Reactive aggression and ODD symptoms showed a more diverse distribution, but were positively skewed. Below, we report the best fitting multi-level models on group differences (see Appendix A for the fit indices of all models).

The level of ODD symptoms, as reported by the parents, was higher in children with DLD compared to children without DLD (*B* = 0.11, 95%CI [0.003, 0.215]). A decline in ODD symptoms was found in older children in both groups (*B* = −0.04, 95%CI [−0.061, −0.009]). Gender, SES, and PIQ did not provide better model fits and were excluded. Reactive aggression did not differ in children with and without DLD. Children in both groups reported lower levels of reactive aggression when they were older (*B* = −0.021, 95%CI [−0.039, −0.002] and girls reported lower levels than boys (*B* = −0.077, 95%CI [−0.151, −0.003]. SES and PIQ did not contribute to the model. The level of proactive aggression also did not differ in children with and without DLD. In both groups, a decline with age was found (*B* = −0.013, 95%CI [−0.021, − 0.005]). Girls reported lower levels of proactive aggression than boys (*B* = −0.030, 95%CI [−0.058, −0.001]) and children with higher PIQ reported lower levels of proactive aggression in both groups (*B* = −0.001, 95%CI [−0.002, −0.000]). SES did not contribute to the model.

Emotion recognition as reported by the parents was lower in children with DLD than in children without DLD (*B* = −0.336, 95%CI [−0.467, −0.207]). An increase was found in older children in both groups (*B* = 0.040, 95%CI [0.005, 0.074]). Anger dysregulation did not differ in children with and without DLD and showed decreasing levels in older children (*B* = −0.073, 95%CI [−0.116, −0.030]). PIQ, SES, nor gender affected these results. Children with DLD had more general (*t*(214):−21.33, *p* < 0.001, *d*:−2.93), pragmatic (*t*(214):−17.76, *p* < 0.001, *d*:−2.44), and emotion communication problems (*t*(136.62):−8.23, *p* < 0.001, *d*:−1.15) compared to children without DLD.

In summary, we found higher levels of ODD symptoms in children with DLD compared to children without DLD, whereas proactive and reactive aggression were similar in both groups. The three externalizing problems decreased as children became older. Emotion recognition was lower in children with DLD, whereas anger dysregulation did not differ between groups. Emotion recognition increased and anger dysregulation decreased as children became older.

### 3.3. Longitudinal Relationships between Emotional Competence and Externalizing Problems

We examined whether individual differences in externalizing problems were longitudinally related to children’s level and development of emotional competence (see Appendix A for correlations between all study variables). The level of ODD symptoms was longitudinally related to emotion recognition and anger dysregulation. Higher mean levels of emotion recognition (*B* = −0.198, 95%CI [−0.286, −0.123]) as well as within-person growth in emotion recognition (*B* = −0.119, 95%CI [−0.189, −0.050]) were related to fewer ODD symptoms as reported by the parents. However, when anger dysregulation was added to the model, the mean level of emotion recognition was not significant anymore (Table 3). The mean level of anger dysregulation was related to more ODD symptoms in both groups, which relationship was stronger in children with DLD as is indicated by the significant interaction effect (Figure 3). Additionally, longitudinal decreases in anger dysregulation within children were related to decreasing levels of ODD symptoms in both groups. When anger dysregulation was added to the model, the difference in the level of ODD symptoms between children with and without DLD was no longer significant, suggesting that problems in anger dysregulation explains the higher levels of ODD symptoms in children with DLD.

Reactive aggression was not related to emotion recognition (mean or change). The mean level of anger dysregulation was related to higher levels of reported reactive aggression, but only in children with DLD, as indicated by the significant interaction effect (Figure 3). Changes across time within individuals in anger dysregulation did not contribute to changes in reactive aggression (Table 3). Additionally, proactive aggression was unrelated to emotion recognition and anger dysregulation after bootstrapping.

In summary, the findings show that children with DLD who have more difficulties regulating their anger have higher levels of ODD symptoms and reactive aggression, but not proactive aggression. In children without DLD, anger dysregulation was also related to more ODD symptoms, but less strongly than in the DLD group and not to both types of aggression. Better emotion recognition was related to more ODD symptoms, but not to aggression, in both groups. Additionally, increasing levels of emotional competence (more emotion recognition and less anger dysregulation) across time, related to lower levels of ODD symptoms in both groups but not to their level of aggression.

### 3.4. Mediating Role of Emotional Competence in the Relationship of Communication Problems and Externalizing Problems

We examined whether emotional competence mediated the relationship between communication problems and externalizing problems (Figure 1). The relationships between communication problems and externalizing problems were considered (c). Next, the relationships between communication problems and emotional competence were examined (a), where after the mediating role of emotional competence in the relationship between communication problems and externalizing problems was examined (c’).

#### 3.4.1. The Relationship between Communication Problems and Externalizing Problems

ODD symptoms were related to more pragmatic (*B* = 0.012, 95%CI [0.006, 0.018]) and emotion communication problems (*B* = 0.264, 95%CI [0.118, 0.391]) in both groups. Reactive aggression was higher in children with more emotion communication problems in both groups (*B* = 0.092, 95%CI [0.006, 0.178]) and in children with DLD with more general (*B* = 0.009, 95%CI [0.003, 0.015]), or pragmatic communication problems (*B* = 0.015, 95%CI [0.003, 0.026]), whereas no relationships were found for children without DLD (*B* = −0.001, 95%CI [-0.004, 0.001]; *B* = −0.001, 95%CI [−0.008, 0.004] respectively). When the CCC scales (speech, syntax, semantics, coherence, pragmatics) were examined separately in children with DLD, only semantic and pragmatic problems related to more reactive aggression. Proactive aggression was positively related to more emotion communication problems in both groups (*B* = 0.043, 95%CI [0.014, 0.073]). Additionally, proactive aggression was related to more general communication problems, but only in children with DLD (*B* = 0.002, 95%CI [0.000, 0.005]), but when the separate CCC scales were examined in children with DLD, none of them reached significance, suggesting that these relationships were not strong.

#### 3.4.2. The Relationship between Communication Problems and Emotional Competence

Emotion recognition was related to fewer general, pragmatic, and emotion communication problems in both groups (*B* = −0.012, 95%CI [−0.016, −0.007]; *B* = −0.028, 95%CI [−0.035, −0.020]; *B* = −0.484, 95%CI [−0.626, 0.341] respectively). When the structural language scales were examined separately in children with DLD, none of them were significant, whereas the pragmatic scale was (*B* = −0.035, 95%CI [−0.048, −0.022]). Anger dysregulation was related to more general, pragmatic and emotion communication problems in both groups (*B* = 0.004, 95%CI [0.001, 0.008]; *B* = 0.012 95%CI [0.005, 0.019]; *B* = 0.371, 95%CI [0.212, 0.530] respectively). However, when the structural language scales were examined separately in children with DLD, none of them reached significance, although the syntactic and pragmatic scales almost reached significance (*p* = 0.083; *p* = 0.077).

#### 3.4.3. Emotional Competence as a Mediator between Communication Problems and Externalizing Problems

Mediation was not examined in proactive aggression, because it was unrelated to emotional competence. The level of ODD symptoms was related to more pragmatic and emotion communication problems and to the two indices for emotional competence in both groups. However, when the communicative and emotional factors were combined, communication problems did not add to the model in addition to the indices of emotional competence, suggesting mediation. A direct test of mediation indicated that the relationships between pragmatic problems and emotion communication problems and ODD symptoms were mediated by the mean level of emotion recognition (*B* = −0.035, 95%CI [−0.307, −0.080]; *B* = −0.033, 95%CI [−0.094, −0.020] respectively). The results showed that more communication problems were related to lower levels of emotion recognition, which in turn related to higher levels of ODD symptoms in children with and without DLD.

Anger dysregulation was not a mediating factor of the relationship between communication problems and ODD symptoms when both groups were examined together. However, because anger dysregulation was more strongly related to ODD symptoms in children with DLD, we also performed this test of mediation for the DLD group alone. Within the DLD group, increasing levels of anger dysregulation across time mediated the relationship between pragmatic problems (*B* = 0.007, 95%CI [0.002, 0.024]), or emotion communication problems (*B* = 0.129, 95%CI [0.065, 0.342]) and ODD symptoms. The results showed that lower levels of communication problems were related to decreasing levels of anger dysregulation across time, which in turn related to decreasing ODD symptoms in children with DLD.

Reactive aggression was related to more communication problems (semantic problems and pragmatic problems) and more mean anger dysregulation in children with DLD. Therefore, mediation was only tested in children with DLD. Semantic problems were related to reactive aggression in addition to anger dysregulation (*B* = 0.057, 95%CI [0.016, 0.089]) and anger dysregulation did not mediate the relationship between semantic problems and reactive aggression. However, the relationship between pragmatic problems and reactive aggression was mediated by changes in anger dysregulation across time (*B* = −0.119, 95%CI [−0.484, −0.0002). Children with DLD with lower levels of pragmatic problems had decreasing levels of anger dysregulation across time, which related to decreasing reactive aggression.

## 4. Discussion

In the present study, different types of externalizing problems were examined longitudinally in children with and without DLD. We differentiated between proactive goal-directed externalizing behaviors and anger-induced reactive externalizing behaviors. Children with and without DLD reported similar levels of proactive aggression. This is in line with earlier findings that children with DLD did not report more bullying, rule-breaking, or delinquent behavior than their peers [4,6,22,23]. Reactive externalizing behaviors include both aggressive and non-aggressive behaviors after goal-thwarting or provocation [19]. The results of our study suggest that children with DLD do not show elevated levels of reactive aggressive behaviors, but do show more non-aggressive externalizing problems such as oppositional behavior.

As can be expected in this age range [20], the level of externalizing problems, especially proactive aggression, was low in both groups and further decreased with age. Thus, as a cohort, the children were following typical developmental trajectories. However, there were marked individual differences within both groups at the level and in the development of externalizing problems across time. We aimed to explain these individual differences between children with and without DLD and within children across time by examining the longitudinal relationships with emotional competence and the severity of communication problems.

In line with our expectations, we found that children’s emotional competence was unrelated to their level of proactive aggression [19,37], whereas emotion communication problems did explain higher levels of proactive aggression in children with and without DLD [26,38]. Children who experience problems communicating about their emotions, may find it more difficult to express their wishes and emotions in a constructive manner and rely more on aggressive means to make themselves known or try to obtain their goals [26,38]. For reactive externalizing problems, emotional competence was a protective factor in children with and without DLD, especially in children with DLD. Moreover, more pragmatic and emotion communication problems related to more reactive externalizing problems, and these relationships were mediated by children’s emotional competence in the DLD group. These results are discussed in more detail below.

### 4.1. Relationships between Emotional Competence and Reactive Externalizing Problems

We found longitudinal relationships between the indices for emotional competence and the two indices of reactive externalizing problems (ODD symptoms and reactive aggression). Children who showed an increasing ability to regulate their anger and to recognize others emotions showed fewer oppositional behaviors across time according to their parents. This suggests that increasing emotional competence can be protective against the development of reactive externalizing problems. Oppositional behaviors may often be caused by misunderstandings, unmet expectations or insecurities. When children do not fully understand what is going on in a social situation or do not feel understood, this causes a negative affect and more oppositional behavior [19,30]. Children gain knowledge about the intentions and emotions of others through social interaction. As children become older, they learn to recognize and understand the increasingly subtle and complex emotions of others and learn to express their own wishes in constructive manners [32,33]. When children gain a better understanding of social interactions and their own role in these interactions, this may diminish negative interactions between children and their peers; and between children and their care-givers. Therefore, it is important to help children to develop their emotional competencies.

A growing body of intervention studies shows that children can gain emotional competencies. Through parent-support interventions, the emotion socialization behaviors of parents can be improved, which has a positive effect on children’s emotional competencies and decreases psychosocial problems [54]. The conversations parents have with their children about emotions, intentions and behaviors can help children to gain a better understanding of emotions, especially when children are active contributors of these conversations [55,56,57]. When children are able to express their ideas and emotions, caregivers are able to explain, elaborate or contradict the interpretations of children and support children in regulating and expressing their emotions. Especially when parents not only label emotions but explain the causes and consequences of emotions, children are able to increasingly gain an emotional understanding [33,55], which is necessary to deal with emotional situations [58,59].

The effects of emotional competence may be more salient in children with DLD. Our results indicated that children with lower levels of anger dysregulation had fewer ODD symptoms, which relationship was stronger in children with DLD. Lower anger dysregulation was also related to lower levels of reactive aggression, but only in children with DLD. Although anger dysregulation is thought to be a risk factor for the development of reactive aggression, this relationship is specifically found in clinical samples [30], whereas no or weaker relationships are reported in community samples [31,60], which is in line with our findings. In typically developing children, the relationship may be less strong because other factors are likely to play a role during development. For instance, negative peer interactions mediate the relationship between emotion dysregulation and reactive aggression in typically developing children [37]. Children with DLD are more vulnerable in social interactions, especially when they experience more problems in emotional competence. Children with DLD with more problems understanding their own and others’ emotions are more often victimized and have lower friendship quality [23,32]. The combination of multiple risk factors in communication, emotional competence, and social relationships, make children more vulnerable for externalizing problems. This is not a simple sum of risk factors, because different problem areas are likely to interact [61]. For instance, children who have problems regulating emotions are less popular with other children. When children have fewer positive interactions with peers, this diminishes learning opportunities to gain an understanding of others’ emotions and intentions and learn to communicate emotions constructively [32,62]. This in turn makes children more vulnerable for negative interactions with peers and externalizing problems. Therefore, problems in emotional competence may have a stronger impact on the development of children with DLD through its interaction with other risk factors.

### 4.2. Emotional Competence as a Mediator between Communication and Externalizing Problems

In addition to emotional competence, the communication problems of children with and without DLD were related to their reactive externalizing problems. Lower levels of pragmatic and emotion communication problems were related to fewer ODD symptoms in children with and without DLD. Additionally, more semantic and pragmatic problems were related to more reactive aggression in children with DLD. However, these relations were partially mediated by children’s emotional competence. These findings are in line with our expectations that communication problems can impede the development of emotional competence, which in turn can put a child at risk for more reactive externalizing problems [9]. However, in addition to emotional competencies, the language areas which are important to understand the meaning of what others are saying or what others are implying are also important for reactive externalizing problems. Semantic problems refer to difficulties understanding the meaning of words and sentences, and difficulties in producing meaningful expressions. Pragmatic problems are also important for understanding others, but refer to difficulties understanding the meaning behind the words, such as in figurative speech and in the understanding of non-verbal communication [46]. In social interactions, children have to combine multiple sources of information to understand what is going on. They need linguistic information to understand what others are saying and pragmatic knowledge to understand the meaning in the social context. Additionally, children need to have insight into the perspective of others, read emotional information on the face, in the tone of voice and combine this information with knowledge about the person, the situation and past experiences. Pragmatic, semantic problems and emotional problems make it difficult to understand the intentions of others, which can lead to more hostile interpretations of social situations. Hostile interpretations are in turn an important predictor of reactive aggression [19]. Our findings suggest that both these communicative and emotional problems are risk factors for reactive externalizing problems in children with DLD.

In interventions for children with DLD, we should acknowledge that a combination of linguistic, pragmatic and emotional problems makes children vulnerable for psychopathology. When we only focus on the linguistic problems of children with DLD we miss important consequences of the language problems in their emotional development. At the same time, we should be aware of the possible underlying linguistic vulnerabilities of children with externalizing problems. Because emotional competence may mediate the relations between communication problems and externalizing problems, there may be a group of children where the emotional and behavioral problems are recognized, but language problems go unnoticed [22,63,64]. A recent review study reported 80% comorbidity of language problems and psychopathology in children who received help for their problems. However, language problems were generally not tested and treated in these children [64]. Especially semantic and pragmatic problems are more difficult to recognize and might be misinterpreted as unwillingness or stubbornness, resulting in negative interaction patterns [63].

Moreover, communicative frustration in children may lead to a greater negative affect, while children at the same time gain less experience understanding, regulating, and expressing their emotions in constructive ways, contributing to more externalizing problems. Additionally, when children receive treatment for externalizing problems, it is likely that children with language difficulties will not benefit from these interventions to the same extent as children with more advanced communication skills. For instance, cognitive-behavioral therapy or group-based interventions place large demands on children’s language abilities [65]. Recognizing these language problems, and an awareness of the relationship between language and externalizing problems and the mediating role of emotional competence is crucial to help professionals recognize and treat the underlying problems causing externalizing behavior [7,22]. The children in our study were all diagnosed with DLD at an early age and had received special support to diminish the negative effects of DLD through speech and language therapy, special education, or school counsellors. This may have protected the development of more extreme levels of externalizing problems, as has been found in children with unrecognized language problems [22,63,64]. Therefore, the early identification and intervention in children with communication problems is warranted.

### 4.3. Limitations and Future Directions

Common measure variance may have influenced our results. In the analyses on proactive and reactive aggression we used both child and parent reports, but the analyses on ODD symptoms included only parent reports. This may have artificially inflated the relations. Although this effect is most prominent when questionnaires are similar in topic and formulation, which was not the case in our study [66], future studies should try to gain information from multiple informants and use both observational measures and questionnaires. This may also be beneficial to gain better understanding of the frequency and contexts in which externalizing problems appear, because the behavior of children is likely to be dependent on the context they are in (e.g., at school or at home) [3] and with whom they are interacting (parents, teachers, friends, or other peers).

Additionally, better insight in the underlying mechanisms which may cause problems in emotional competence in children with DLD is necessary to tailor interventions. Children may experience emotion regulation problems because they are more aroused after provocation, or because they experience difficulties to regulate their arousal due to problems in executive functioning [67]. However, children may also become more aroused because they interpreted their surroundings as (too) negative due to a lack of insight in others’ minds and behaviors [19]. A combination of physiological measures and social information processing measures could be used to gain an insight into the possible underlying difficulties in emotion regulation of children with DLD.

Another limitation was that the communication problems were only measured once. Therefore, we were unable to examine the longitudinal effects of changes in the communication problems of children across time. Future studies should not only include longitudinal measures of general, pragmatic, and emotion communication, but ideally start the study in younger ages, as the developments we examined already start early in life. Moreover, developmental changes of children are likely to influence the relationship between emotional competence and externalizing problems. We were unable to examine differences in different age ranges due to power issues, but future studies should examine the interrelation of communicative, emotional, and externalizing problems in more detail. The relation between language problems may be more salient in earlier ages, whereas the role of emotional competence may increase in strength as children become older. A recent review, for instance, found that the relation between language abilities and social problems with peers diminished as children became older [68]. Alternatively, it is possible that emotional competence already mediates the relation between language problems and externalizing problems from an early age, because language is such an important tool in the early emotion socialization process [69,70].

## 5. Conclusions

Although interventions for children with DLD typically focuses on their language problems, our study suggests that the secondary effects of language problems in emotional competence should not be overlooked. Children with DLD were more vulnerable for reactive externalizing problems, especially when they had more problems recognizing emotions and regulating emotions. However, just as in children without DLD, the development of these emotional competencies related to decreasing levels of reactive externalizing problems. Additionally, children who were able to differentiate and communicate about their emotions reported lower levels of proactive aggression. Children use language throughout their lives to interact with others. These social interactions are crucial for social learning [69,71]. When the development of emotional competence is delayed as a consequence of DLD, problems in emotional competence are likely to further diminish the chances for children to experience positive social interactions with others and learn from these experiences [32,72]. In interventions, we should ask ourselves which learning opportunities children are missing to communicate, to learn about emotions and develop positive social relationships, so that we are able to focus our attention on those developments, parallel with language interventions. 

## Figures and Tables

**Figure 1 ijerph-17-06008-f001:**
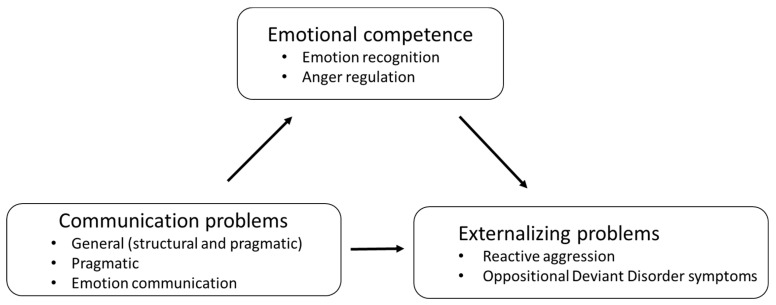
Emotional competence as a mediator between communication problems and reactive externalizing problems.

**Figure 2 ijerph-17-06008-f002:**
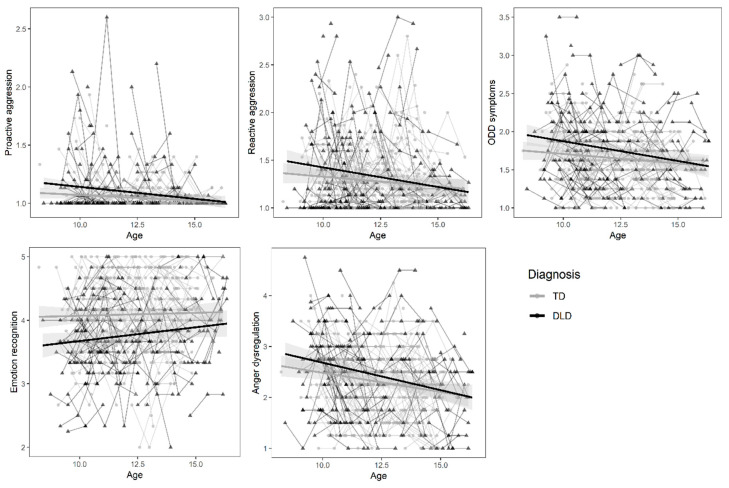
Representation of the raw data for proactive aggression, reactive aggression, Oppositional Deviant Disorder (ODD) symptoms, emotion recognition and anger dysregulation. The regression lines represent the mean level in children with Developmental Language Disorder (DLD) and typically developing children (TD) across time with 95% confidence interval.

**Figure 3 ijerph-17-06008-f003:**
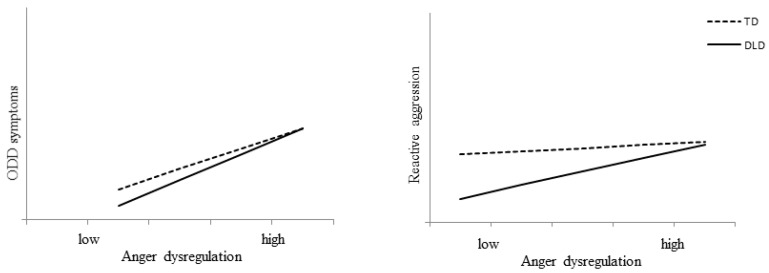
Moderation effect of diagnosis on the longitudinal relation between mean anger dysregulation and externalizing problems for children with Developmental Language Disorder (DLD) and typically developing children (TD).

**Table 1 ijerph-17-06008-t001:** Characteristics of participants at Time 1 for children with and without Developmental Language Disorder (DLD).

	With DLD	Without DLD
Number of children—*n*	98	156
Male	47 (48.0%)	68 (43.6%)
Female	51 (52.0%)	88 (56.4%)
Mean Age in years (*SD*)	11.5 (1.1)	11.6 (1.4)
Age range in years, months	9.2−16.3	9.8–15.4
Neighborhood socio-economic status (SES) (*SD*) ***	0.00 (1.10)	0.66 (1.12)
Range neighborhood SES	−4.19–2.50	−5.24–2.44
Performance IQ (PIQ)—*n*	92	146
PIQ (*SD*) ***	93.8 (13.1)	109.4 (17.1)
Range PIQ	70−140	78−140

Note: The neighborhood SES of the participating parents is determined by the mean level of education, occupation, and income of all adults of a neighborhood compared to the other neighborhoods in the Netherlands (M = 0.28, SD = 1.09, Range = −6.8 to 3.1); *** *p* < 0.001.

**Table 2 ijerph-17-06008-t002:** Psychometric properties of the questionnaires for children with and without Developmental Language Disorder (DLD).

	Range	*N*	*α* Time 1		Grand Means (*SD*)	
		Items	With DLD	Without DLD	With DLD *n* = 98	Without DLD *n* = 156
**Externalizing problems**						
Oppositional Deviant Disorder symptoms	1–4	8	0.89	0.79	1.76 (0.49)	1.65 (0.34)
Reactive aggression	1–3	15	0.96	0.89	1.33 (0.40)	1.28 (0.24)
Proactive aggression	1–3	15	0.91	0.91	1.09 (0.17)	1.04 (0.07)
**Emotional competence**						
Emotion recognition	1–5	6	0.75	0.75	3.77 (0.56)	4.08 (0.53)
Anger dysregulation	1–5	4	0.81	0.75	2.45 (0.74)	2.30 (0.56)
**Communication problems**						
Emotion	1–4	14	0.91	0.91	2.03 (0.58)	1.43 (0.42)
General	53–157	56	0.83	0.87	115.52 (13.65)	73.33 (15.01)
Pragmatic	24–78	28	0.83	0.79	54.92 (7.46)	35.97 (7.94)
Speech	8–24	7	0.75		16.10 (3.54)	
Syntax	7–20	7	0.59		15.29 (2.48)	
Semantics	5–18	7	0.69		14.21 (1.74)	
Coherence	6–20	7	0.80		15.00 (2.38)	

Note: *N* Emotion communication problems (CAM): DLD = 87, without DLD = 151; General communication problems (CCC) scales: DLD = 92, without DLD = 142.

**Table 3 ijerph-17-06008-t003:** Regression weights with 95%CI for best-fitting models with emotion recognition and anger dysregulation predicting reactive aggression and Oppositional Deviant Disorder (ODD) symptoms.

		Reactive Aggression	ODD Symptoms
Age		−0.021 [−0.031, 0.008]	−0.007 [−0.024, 0.010]
Gender		**−0.073 [−0.149, −0.005]**	-
Diagnosis		**−0.404 [−0.707, −0.057]**	−0.207 [−0.436, 0.021]
Emotion recognition	Mean	-	−0.049 [−0.112, 0.013]
	Change	-	**−0.108 [−0.177, −0.038]**
Anger dysregulation	Mean	0.041 [−0.014, 0.118]	**0.384 [0.312, 0.455]**
	Change	0.018 [−0.052, 0.094]	**0.133 [0.060, 0.207]**
Diagnosis x Anger dysregulation	Mean	**0.182 [0.025, 0.312]**	**0.105 [0.003, 0.208]**
	Change	0.090 [−0.037, 0.230]	0.092 [−0.044, 0.229]

Note: Significant regression weights are bold.

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
