# Peer review of "Emotional Competence Mediates the Relationship between Communication Problems and Reactive Externalizing Problems in Children with and without Developmental Language Disorder: A Longitudinal Study"

_ijerph, 2020, doi:10.3390/ijerph17166008_

Round 1

Reviewer 1 Report

  1. In Figure 1, "Pragmatic" appears under General and as a category by itself. Did the authors put Pragmatic here twice on purpose?
  2. In the Introduction section, the authors mentioned “The communication problems of children with DLD are not explained by other neurodevelopmental disorders, hearing loss, or intellectual disabilities," but in the Materials and Methods section, only “no diagnosis of autism spectrum disorder or hearing loss” was mentioned. Were other neurodevelopmental disorders and intellectual disabilities exclusion criteria in this study?
  3. Were the questionnaires filled out only by parents, not by nannies, grandparents, or other primary caregivers?
  4. Were the parents and children monolingual or bilingual speakers? Was Dutch the primary language of instruction in school?
  5. When did the parents fill out the questionnaires at the three time points across the 18 months? The first month, ninth months, and eighteenth months?
  6. It would be helpful to present the data collection process in a flow chart, such as the scales used at each time point, the number of participants excluded for certain reasons.
  7. It would be helpful to present the results with some figures. For example, is it possible to show “Multi-level analyses showed that increasing emotion recognition and decreasing anger dysregulation were longitudinally related to decreasing ODD symptoms in both groups,…“ in a figure?
  8. Some numbers were not aligned in Table 3.
  9. The results of the study are intriguing, but is it possible that the age ranges of the participants were too wide? Children’s development changes significantly within several years, so would the results be different if you analyze the data in two groups, such as 8-12 and 13-16-year-old groups?
  10. A possible future direction could be to include data from observing children’s behavior (e.g., frequency of occurrence) in addition to having survey answers.

Author Response

Thank you for reviewing our article. Please see the attachment. 

Reviewer 2 Report

Thank you for submitting this paper for review.  It is of interest and provides valuable longitudinal data which you highlight is missing from the evidence base.

Unfortunately your paper has some serious flaws that must be addressed before it can be recommended for publication and I will discuss these by section. One thing I would suggest upfront is that you have a native English speaker read your work and make the necessary style changes regarding words and phrases.  Your second author would be an ideal candidate to do this for you, as there are numerous errors that need fixing.

Introduction:

You present a lot of information here and describe the work that has been done previously.  This section mainly needs work in style, English language and grammar.  You provide a clear description of your aims and hypotheses which you refer back to. 

Method:

Over all your methods need extensive revision.  I have found considerable presentation of results in the methods and these need to be removed.  The design section is far too small.  it does not adequately describe the larger study, simply referencing it is on enough.  You do not mention when the time points are which makes it difficult to understand if there was a base line or simply a six month window between data collection.  You do not adequately describe the participants.  parents would also be participants as they filled in the questionnaires on the child's behalf. Did you obtain consent from them?  You do not provide information on how you obtained active consent.  Did you use a validated protocol? Did you obtain consent at each data collection time point? You do not describe why there is attrition from children; did they move? Was the trial too onerous?

Results:

The results section needs to include the information from the methods. There is some discussion of the results in this section, so I suggest looking this over. The “these findings suggest” paragraph is one that could be moved or omitted from the results section.

Relations is the wrong word. You need to use the word relationship or relationships.

You provide so much information in the results (relationships you’ve tested), it would be useful to have some kind of matrix with all the relationships that you are testing and what you found to be significant. This would give the reader an easy overview of your findings.

Discussion:

The discussion is a little like a restating of the results. There needs to be much more of a discussion of your findings with better linking of your findings to the literature.

In your limitation section you mention that the communication problems were only measured once, but this is not identified in the methods. You need to describe why it is only measured once and justify this in the methods before you can identify this in the discussion as a limitation.

Also there is little ‘future direction’ rather a sentence about future research. I want to know about why this research is important and what it can change. What is the gap you have filled that can now fix, resolve or be used to develop something new to support children and families? There should be a bigger picture here about impact you can foresee from this research.

Conclusion:

The last sentence is confusing since you haven’t mentioned interventions once in this paper. I suggest removing. You conclusion does not highlight anything in relation to your work/results. Your conclusion should restate the issue as you’ve done, but then say that now we have this new information ( from your study) we can now look to do this…

Author Response

(The authors gave the same response as above.)

Round 2

Reviewer 2 Report

Thank you for your revisions to this paper.  I think it is now significantly improved. Well done.